# Blinded by the Mind: Exploring the Hidden Psychiatric Burden in Glaucoma Patients

**DOI:** 10.3390/biomedicines13010116

**Published:** 2025-01-07

**Authors:** Jeniffer Jesus, João Ambrósio, Dália Meira, Ignácio Rodriguez-Uña, João Melo Beirão

**Affiliations:** 1Unidade Local de Saúde de Entre o Douro e Vouga, 4520-211 Santa Maria da Feira, Portugal; alvesambrosio.joao@gmail.com; 2Unidade Local de Saúde de Gaia e Espinho, 4434-502 Vila Nova de Gaia, Portugal; 3Instituto Oftalmológico Fernández-Vega, 33012 Oviedo, Spain; 4Unidade Local de Saúde de Santo António, 4099-001 Porto, Portugal

**Keywords:** glaucoma, anxiety, vision impairment, depression, psychiatric burden, mind, permanent vision loss, quality of life, mental health challenges, sleep problems

## Abstract

Glaucoma is one of the leading causes of permanent vision loss worldwide and has a profound impact on patients’ quality of life. Vision impairment is strongly associated with several psychiatric disorders, like depression, anxiety, and sleep problems. These psychiatric issues are often exacerbated by the gradual, irreversible, and typically silent progression of the disease, contributing to increased mental health challenges for affected individuals. A systematic review was conducted following PRISMA guidelines across six different databases (CINAHL, MEDLINE, PsycINFO, Web of Science, Scopus, and the Cochrane Library) and one gray literature source (Google Scholar), covering the period from 2013 to 2024. Twenty-nine studies involving a total of 13,326,845 subjects were included in the synthesis, highlighting a considerable prevalence of psychiatric disorders among glaucoma patients. Depression and anxiety were the most common conditions identified, with depression rates ranging from 6.6% to 57% and anxiety from 12.11% to 49%. Other less frequent but still significant conditions like sleep disorders, psychosis, dementia, and post-traumatic stress disorder were also observed. The findings also indicated that psychiatric severity was influenced by socio-demographic factors, glaucoma severity, and treatment duration. Given the high occurrence of psychiatric pathologies among individuals with glaucoma, it is essential to develop comprehensive care strategies that address both eye and mental health needs. Multidisciplinary collaboration among ophthalmologists, psychiatrists, psychologists, and primary care physicians is crucial for developing personalized treatment plans that effectively manage both the ocular and psychological aspects of the disease.

## 1. Introduction

Glaucoma is a major cause of permanent vision loss worldwide, currently affecting more than 76 million individuals, a number expected to rise to 111.8 million by the year 2040 [1]. Chronic illnesses are extensively documented to have a detrimental effect on mental health, with conditions like diabetes, cardiovascular diseases, and cancer often linked to higher rates of depression and anxiety [2]. As a chronic disease, the correlation between glaucoma and psychological illnesses is complex and multifaceted. The disease not only causes vision impairment but also has a severe impact on patients’ quality of life and presents considerable socioeconomic issues. The psychological effects experienced by people suffering from glaucoma are significant, occurring alongside the physical symptoms of the condition [3,4]. Impaired vision can lead to a deep sense of reliance, diminished independence, and reduced social engagement [5]. Researchers suggests that individuals with visual impairments are nearly twice as likely to experience depression compared to those without visual problems [6]. These conditions can be exacerbated by the challenges experienced by people with glaucoma, impacting their quality of life, adherence to therapy, and general outlook [7,8]. The gradual and often silent progression of glaucoma can result in notable tension and anxiety, primarily due to the fear of eventual loss of vision. Studies have indicated that around 40% of individuals diagnosed with glaucoma suffer from anxiety, and approximately 20% experience depression rates that are significantly higher than those observed in the general population [9].

Despite the extensive documentation of psychiatric issues in other chronic conditions, the relationship between glaucoma and psychiatric disorders has not been fully elucidated, leaving a gap in the current literature. Existing reviews, while insightful, have not consistently explored the potential mechanisms linking glaucoma and mental health disturbances. There is a need to delve deeper into how glaucoma could affect the brain areas associated with mood regulation, stress responses, and psychological resilience. This exploration is crucial for uncovering not only the prevalence of these psychiatric issues but also the underlying pathways that may drive these associations.

To fully understand the extent of psychiatric disorders in individuals with glaucoma, it is essential to recognize the reciprocal connection between these conditions. Emerging research suggests that glaucoma may not only induce psychological disorders due to the visual impairment it causes but may also be influenced by psychiatric conditions that exacerbate its progression, creating a cycle of physical and mental decline. Additionally, brain regions involved in both vision and emotional regulation, such as the occipital cortex and limbic system, may be implicated in this relationship, further complicating the disease’s impact on mental health. Previous research on this topic has been dispersed, exhibiting differences in their approaches, sample sizes, and geographical scopes. Understanding the interaction between glaucoma and psychological problems is essential for developing comprehensive treatment strategies. By combining mental health care with eye care, treatment adherence can be improved, quality of life can be enhanced, and the disease’s progression can potentially be slowed [10].

Our systematic review aims to analyze and evaluate the current body of research on psychological disorders in individuals with glaucoma, gathering and summarizing the existing research, providing a comprehensive examination of the frequency, categories, and consequences of mental problems in people with glaucoma. In addition, we also aim to contribute new insights regarding the biological pathways that may link psychological stress and glaucoma progression, emphasizing the need for a more integrative approach in research and clinical practice. Specifically, the main objectives of the study are (1) to assess the prevalence of psychological conditions in individuals diagnosed with glaucoma; (2) to classify the psychiatric illnesses frequently linked to glaucoma; (3) to explore possible mechanisms linking glaucoma progression to brain areas involved in mood regulation and how stress-related hormonal changes may exacerbate glaucoma; (4) to analyze the impact of various mental health conditions on the treatment of glaucoma and patient outcomes; and (5) to identify gaps in the current body of knowledge and suggest potential areas for future research.

Through this investigation, our goal is to enhance the understanding of the difficulties experienced by individuals with glaucoma and encourage the implementation of more efficient and coordinated care approaches. Additionally, we aim to identify notable trends and discrepancies in the prevalence and impact of psychiatric disorders among glaucoma patients by analyzing newer studies from Africa, Asia, Europe, and North America and from underrepresented regions, and with unique socio-demographic contexts.

Therefore, this review will be an important resource for clinicians, academics, and policymakers, highlighting the importance of a multidisciplinary approach for managing glaucoma that includes mental health support.

## 2. Materials and Methods

This systematic review was conducted according to the PRISMA 2020 Guidelines and registered the systematic review project via Open Science Framework (OSF) with the following site doi: 10.17605/OSF.IO/WP7J8.

### 2.1. Search Strategy

The search strategy for this systematic literature review was designed to capture a wide range of research on psychiatric pathologies in the glaucoma population. Searches were conducted and reviewed independently by three authors (initials of authors: J.J., J.A. and J.M.B.), ensuring comprehensive coverage. In cases of disagreement, a fourth author (D.M./I.R.) resolved the conflicts. Searches were comprehensively conducted across six electronic databases: CINAHL, MEDLINE, PsycINFO, Web of Science, Scopus, and the Cochrane Library. Additionally, to minimize publication bias, the articles search was comprehensively conducted in Google Scholar to trace and review both scholarly sources and gray literature [11]. The search strategy included articles published since 2013. The Medical Subject Headings (MeSH) search terms used were “glaucoma”, “psychiatric disorders”, “mental health”, “depression”, “anxiety”, and “psychological distress”. Boolean operators (AND, OR) were utilized to refine and expand the search as necessary. Additionally, we also reviewed the reference lists of relevant articles and conducted manual searches of key journals to identify any studies that might have been missed in the database searches. To ensure that the selection process was transparent and reproducible, duplicate records were removed using reference management software, and a two-step screening process was implemented: (1) titles and abstracts were screened for relevance, and (2) full texts of potentially relevant studies were reviewed in detail against the inclusion and exclusion criteria. The literature selection was represented in PRISMA model at Figure 1 and the relevant selected articles are explained in the data extraction table—Table 1. After the data collection process, a meta-analysis of all the selected studies was performed using the IBM SPSS version 30 software.

### 2.2. Inclusion/Exclusion Criteria

The inclusion criteria for this systematic literature review encompassed studies published between 2013 and 2024 that investigated the prevalence, incidence, or impact of psychiatric pathologies, such as depression, anxiety, or psychological distress, in populations diagnosed with glaucoma. Eligible studies included various study designs, such as randomized controlled trials, cohort studies, case–control studies, and cross-sectional studies. Participants needed to have a clinical diagnosis of any type of glaucoma (e.g., primary open-angle glaucoma, primary angle-closure glaucoma). Studies published in English were included, with no restrictions on geographic location or healthcare settings. Exclusion criteria were defined with particular care to avoid confounding factors that could impact the results. Studies focusing on populations with significant comorbidities, including cardiovascular diseases, metabolic disorders (e.g., diabetes), and severe neurological disorders, were excluded. This decision was made to ensure that any observed psychiatric symptoms were primarily attributable to glaucoma and not to other chronic or progressive conditions. To ensure the reliability of the results, studies that did not assess psychiatric outcomes, case reports, editorials, commentaries, and studies with insufficient methodological quality or data were excluded.

### 2.3. Risk of Bias

In this systematic review, a comprehensive risk of bias assessment was conducted for all included studies to ensure the validity and reliability of the findings. A total of 29 studies were evaluated using appropriate risk of bias tools based on study design. Specifically, the Risk of Bias in Non-randomized Studies of Interventions (ROBINS-I) tool was applied to six cohort studies. The ROBINS-I tool evaluates risk of bias across seven domains: confounding factors, selection of participants, classification of interventions, deviations from intended interventions, missing data, measurement of outcomes, and selection of the reported result, providing a comprehensive assessment of potential biases in non-randomized studies.

For nine systematic reviews included in the analysis, the Risk of Bias In Systematic Reviews (ROBIS) tool was used. This tool assesses bias across four phases: assessing relevance, identifying concerns with the review process (study identification, selection, data collection, and synthesis), and judging the overall risk of bias in the context of the systematic review methodology.

For the remaining 14 cross-sectional and case–control studies, the Newcastle–Ottawa Scale (NOS) was employed. This scale evaluates risk of bias across three broad categories: selection (representativeness of cases/controls, selection of controls, and exposure ascertainment), comparability (controlling for key confounders), and outcome/exposure assessment. Each category is rated to determine overall study quality and potential sources of bias.

## 3. Results

The 29 studies included in this study focus on diverse populations, including glaucoma patients with various disease severities, veterans, and elderly individuals (n = 13,326,845). They encompass a range of geographic regions (e.g., Japan, Taiwan, USA, Europe, and Ethiopia), employing diverse study designs, including cross-sectional, case–control, cohort, and systematic review designs. The most frequently studied psychiatric disorders were anxiety, depression, PTSD, sleep disorders, cognitive impairment, and common mental disorders (CMDs).

Figure 2 shows the meta-analysis forest plot depicting the effect sizes of multiple studies included in a meta-analysis, each contributing to the overall estimated effect size. The confidence intervals of individual studies vary, indicating less certainty in their estimates. The estimated overall effect size is around 2.39 with a confidence interval that does not cross zero, suggesting a statistically significant effect. Additionally, the heterogeneity results show substantial variability among the studies included in this meta-analysis. The random-effects model used accounts for this heterogeneity, implying that the effect sizes differ across studies beyond random chance, possibly because of the differences in study populations or methodologies.

The bubble plot in Figure 3, using effect size as a moderator, visually represents the relationship between effect sizes across primary studies included in the meta-regression. The plot shows a positive slope in the regression line, indicating a potential increase in effect size across studies. When larger effect sizes occur, there is considerable variability in these estimates.

The studies utilized diverse assessment instruments and procedures to evaluate psychiatric diseases in individuals with glaucoma. The commonly used assessment tools included the Hospital Anxiety and Depression Scale (HADS) [39], the Beck Depression Inventory (BDI) [40], the Generalized Anxiety Disorder 7-item (GAD-7) scale [41], and the Pittsburgh Sleep Quality Index (PSQI) [42].

The risk of bias assessment for all the 29 studies included in this systematic review was performed using three tools tailored to the study designs: ROBINS-I (Risk of Bias In Non-randomized Studies of Interventions) for six cohort studies, ROBIS (Risk of Bias in Systematic Reviews) for nine systematic reviews, and the Newcastle–Ottawa Scale (NOS) for 14 cross-sectional and case–control studies.

For the ROBINS-I tool (Figure 4), the cohort studies displayed a predominantly low risk of bias across domains, including confounding factors, participant selection, and deviations from intended interventions. However, moderate risk of bias was observed in some studies concerning the classification of interventions and measurement of outcomes, largely due to reliance on administrative and self-reported data. The traffic light plot that visually represents the overall risk of bias for each study is showed in Figure 5.

The ROBIS assessment for systematic reviews (Figure 6) identified notable areas of bias. While the ‘Study Eligibility Criteria’ domain generally showed low risk, the ‘Identification and Selection of Studies’, ‘Data Collection and Study Appraisal’, and ‘Synthesis and Findings’ domains demonstrated a high to critical risk of bias. This was attributed to inconsistencies in study selection, insufficient data collection protocols, and limitations in synthesizing findings, which may influence the reliability of systematic review conclusions. The traffic light plot corresponding to the risk of bias for each study is showed in Figure 7.

In the evaluation using the Newcastle–Ottawa Scale (NOS) (Figure 8), the cross-sectional and case–control studies were assessed across domains related to selection, comparability, and outcome assessment. The traffic light plot that represents the overall risk of bias for each study is showed in Figure 9.

So overall, while many studies demonstrated strong control of confounding factors and robust data management, issues related to study selection, exposure measurement, and the synthesis of findings were observed, particularly among the systematic reviews and certain cross-sectional studies.

Regarding the analysis of 29 studies conducted in different locations and involving diverse populations, it is evident that there is a significant occurrence of psychological disorders among glaucoma patients. Generally, glaucoma becomes more prevalent with age, affecting an estimated 3.54% of the global population between the ages of 40 and 80 [43].

Depression and anxiety are the most common psychiatric diseases within this group, compared to normal individuals, with depression rates ranging from 6.6% to 57% and anxiety rates from 12.11% to 49% [43].

Depression is commonly identified as a comorbid condition in the retrieved articles analyzed. The meta-analysis conducted by Groff et al. (2023) [9] demonstrated a greater occurrence and intensity of depression among those diagnosed with glaucoma. In their study, Chen et al. (2018) [20] found that glaucoma patients had a considerably higher risk of developing depression compared to the control group. The occurrence of depression showed notable variations in the research. Regarding depression treatments, particularly selective serotonin reuptake inhibitors (SSRIs), Chen et al. (2016, 2017) [16,18], found that individuals with glaucoma were significantly more likely to have been exposed to SSRIs compared to the general population. Additionally, individuals receiving SSRI therapy for Major Depressive Disorder may be at an elevated risk for any type of glaucoma, especially those between the ages of 32 and 70 years, at higher doses (>20 mg) and with long-term use (>365 days). For those who were prescribed SSRIs, there was a 5.80-fold increased risk of acute angle closure glaucoma within the next seven days.

Glaucoma patients frequently report experiencing anxiety. Research conducted by Groff et al. (2023) [9] and Ulhaq et al. (2024) [35] has shown a significant occurrence of anxiety, with combined prevalence rates of 31.2% for symptoms of anxiety and 19.0% for diagnosed anxiety disorders. Furthermore, pediatric glaucoma patients had significantly elevated levels of anxiety (58.6%) compared to adults (29%) [35,44].

Additionally, sleep disturbances are also common among glaucoma patients. Groff et al. (2023) [9] documented a greater incidence of sleep disorders in this group. A study conducted by Ayaki et al. (2014) [12] revealed that those suffering from advanced glaucoma experienced significantly poorer sleep quality in comparison to the control group. Their study also identified a strong association between the severity of glaucoma and sleep quality.

Finally, psychiatric disorders such as psychosis, dementia, and PTSD were investigated less frequently but still documented in several studies. Wändell et al. (2022) [30] discovered a reduced likelihood of primary open-angle glaucoma (POAG) in individuals of both genders who had dementia. Additionally, they observed that women with psychosis exhibited a decreased risk of developing POAG. Stringham et al. (2024) [22] linked PTSD to reduced adherence to glaucoma treatment in military veterans.

Putting all the findings together, our results identify notable trends and discrepancies in the prevalence and impact of psychiatric disorders among glaucoma patients. These data are shown in Table 2. Western populations report higher rates of diagnosed depression and anxiety, while resource-limited regions face greater systemic barriers. Additionally, disease severity universally correlates with worse psychiatric outcomes, though early-stage glaucoma is linked to higher anxiety in younger populations.

## 4. Discussion

### 4.1. The Interplay Between Glaucoma and Mental Health

The review highlights the profound impact of glaucoma on mental health, with findings indicating a significant association between glaucoma and various psychiatric disorders, including depression, anxiety, and sleep disturbances. The underlying mechanisms for this association are multifaceted, involving biological, psychological, and social factors. Factors such as the severity of glaucoma, the duration of treatment, and concomitant visual impairment were identified as key determinants of the prevalence and severity of psychiatric conditions. Additionally, the data showed that stress is a critical factor influencing both psychological well-being and the progression of chronic diseases like glaucoma. The interplay between stress-related hormonal pathways and glaucoma can exacerbate disease progression through mechanisms such as cortisol dysregulation, altered autonomic function, and neuroinflammation [45]. On one hand, chronic stress results in dysregulated cortisol levels, upregulating glucocorticoid receptors in the trabecular meshwork, reducing aqueous humor outflow and raising intraocular pressure (IOP) [46]. Additionally, this impaired homeostasis, and the exacerbation of the disease progression stress leads to a hyperactivation of the sympathetic nervous system, causing increased vascular resistance and reduced perfusion to the optic nerve head, which can worsen optic nerve damage. On the other hand, it contributes to parasympathetic suppression, which affects the body’s ability to recover from stress and maintain normal IOP and vascular homeostasis, and stimulates the production of pro-inflammatory cytokines (e.g., IL-1β, TNF-α), which can impair optic nerve function by increasing glial cell activation and disrupt the blood–retinal barrier, compounding retinal and optic nerve damage [46].

### 4.2. Depression and Antidepressant Use

Patients with glaucoma are at a significantly higher risk of developing depression, reflecting the substantial psychosocial burden associated with visual impairment [5]. Evidence suggests that individuals with glaucoma are more likely to be prescribed selective SSRIs compared to the general population, likely as a reactive approach to managing the psychological impact of the disease. However, many antidepressants, including SSRIs, are contraindicated for glaucoma patients due to their potential to increase IOP, particularly in those predisposed to angle-closure glaucoma [47]. For instance, SSRIs have been linked to a 5.80-fold increase in the risk of acute angle-closure glaucoma within seven days of use in susceptible individuals [47,48,49].

Given this risk, it is imperative to consider newer antidepressants that lack atropine-like side effects. Collaborative care between ophthalmologists and mental health providers is essential to monitor ocular health and ensure safe adjustments to psychiatric medications. Such strategies would help mitigate the risks associated with SSRI use while ensuring effective management of depression [8].

### 4.3. Anxiety Disorders

Anxiety is also significantly more prevalent among individuals with glaucoma compared to healthy individuals. These findings highlight the mental strain glaucoma places on patients, adding to their physical and emotional burden. The higher prevalence of anxiety in younger patients suggests that early diagnosis and the lifelong management of the disease may trigger heightened anxiety in children and adolescents [3,50]. Considering that many anxiolytics are also contraindicated due to their potential to elevate IOP, alternative therapies, such as cognitive–behavioral therapy (CBT), lifestyle changes, and newer medications with lower IOP risks, should be explored [49,51]. These strategies can help reduce reliance on medications that might aggravate glaucoma while effectively managing anxiety symptoms [51].

### 4.4. Sleep Disturbances

Sleep disturbances were also found to be more common among glaucoma patients. These problems could be driven by the physical discomfort and stress related to visual decline, which in turn worsens the patient’s quality of life. Additionally, a strong association was identified between the severity of glaucoma and sleep quality [9,52]. This suggests that as the disease progresses, sleep disturbances may become more prevalent, possibly due to increased anxiety and visual impairment. Addressing these sleep issues with non-pharmacological approaches, such as sleep hygiene techniques and mindfulness practices, could play an essential role in improving both the physical and mental health outcomes of glaucoma patients [51].

### 4.5. Other Psychiatric Conditions

While less frequently studied, conditions such as psychosis, dementia, and post-traumatic stress disorder (PTSD) were also reported in glaucoma patients. The relationship between psychosis and glaucoma may involve complex neurological and pharmacological interactions. Similarly, PTSD may result in decreased treatment adherence, exacerbating visual decline. The potential relationship between glaucoma and dementia raises questions about neurodegenerative processes and their interplay with disease progression [53,54]. These findings highlight the need for specialized mental health interventions tailored to these vulnerable populations.

### 4.6. Socio-Demographic Influence

Socio-demographic factors, including age, gender, and socio-economic status, significantly influence the psychological outcomes of glaucoma patients [55,56]. For instance, studies suggest that women with depression are more likely to develop primary open-angle glaucoma (POAG), potentially due to underlying biological or psychosocial vulnerabilities [30,54]. Interestingly, living in neighborhoods with higher socio-economic status was associated with an increased risk of POAG, potentially due to greater access to healthcare and higher rates of diagnosis. These findings underscore the complexity of socio-demographic influences and highlight the need for targeted interventions addressing the specific needs of different population groups [57]. The prolonged disease trajectory can additionally lead to feelings of hopelessness and frustration, further exacerbating the mental health burden [3,27]. Rezapour et al. (2018) [21] discovered that there was no notable correlation between self-reported glaucoma and depression or anxiety, even after accounting for age and gender. This inconsistency across studies suggests that socio-demographic factors might interact in complex ways with mental health outcomes, necessitating further research to uncover the underlying mechanisms. Our results highlight how geographic, cultural, clinical, and socioeconomic factors influence the prevalence and impact of psychiatric comorbidities in glaucoma patients across Africa, Asia, Europe, and North America, including those from underrepresented regions or with unique socio-demographic contexts not covered by Ramesh et al. or Klugah-Brown et al. By identifying notable trends and discrepancies in the prevalence and impact of psychiatric disorders, these insights underscore the importance of context-specific interventions, such as tailored mental health care models and culturally sensitive education campaigns.

### 4.7. Assessment Tools and Methodologies

The studies reviewed employed diverse methodologies and standardized assessment tools, such as the HADS [39], BDI [40], GAD-7 [41], and PSQI [42]. The use of validated tools ensured reliable data and facilitated cross-study comparisons, enhancing the generalizability of findings. In addition, a wide range of study methodologies were employed, providing methodological diversity and a multi-perspective understanding of how glaucoma impacts mental health, drawing from both observational and experimental data.

Regarding the risk of bias, the results indicated that most studies had a low risk of bias in selection and comparability. Nonetheless, moderate to high risks were noted for some studies due to limitations in outcome ascertainment and exposure assessment. Specifically, the absence of detailed non-response information and inconsistencies in control group definitions contributed to these elevated risks.

### 4.8. Future Directions

Future research should focus on better understanding the link between glaucoma and psychological illnesses. Longitudinal studies are important to uncover causal relationships and to see how psychological symptoms evolve alongside the severity and treatment of glaucoma. It is also important to explore underlying factors, like the role of prolonged stress and vision loss, in contributing to mental health challenges. Research efforts should prioritize evaluating the effectiveness of psychological therapies, support groups, and patient education programs in improving mental health outcomes. Additionally, socio-demographic influences on mental health should be deeply explored to develop targeted interventions for at-risk groups. Our ambition is to expand knowledge and improve care for individuals with glaucoma by addressing both their ocular and mental health in a comprehensive, holistic way [58].

## 5. Conclusions

The goal of this systematic review was to examine the prevalence and types of mental illnesses in individuals diagnosed with glaucoma. The findings reveal that people with glaucoma experience significantly higher rates of psychiatric conditions, particularly depression and anxiety, compared to the general population. Sleep disturbances were also notably more common, with a clear link between the severity of glaucoma and the extent of sleep disruptions. Although less attention was given to conditions such as PTSD and psychosis, they were still reported in some cases. Socio-demographic factors, such as age, gender, and socio-economic status, were also found to influence mental health outcomes in people with glaucoma.

The chronic nature of glaucoma, along with its potential to cause irreversible vision loss, contributes to elevated levels of depression, anxiety, and other psychiatric conditions. The emotional burden of progressive vision loss can impair daily functioning and intensify feelings of isolation and helplessness in patients. The results underscore the urgent need for comprehensive care strategies and interdisciplinary collaboration to address the mental health needs of glaucoma patients. Ophthalmologists must remain vigilant for signs of depression, anxiety, and other mental health concerns in their patients. Routine mental health assessments are recommended throughout the disease course, especially during times of treatment adjustments or disease progression. The timely detection and management of mental health conditions in glaucoma patients can significantly enhance their quality of life and adherence to treatment. Developing shared care protocols and guidelines that address both vision loss and mental health is crucial for integrated care. The screen of high-risk groups should prioritize mental health assessments for advanced-stage glaucoma patients, as they are at higher risk for depression and anxiety, as well as younger patients diagnosed with glaucoma, who often exhibit heightened anxiety related to lifelong disease management, and patients experiencing sleep disturbances, which are commonly associated with advanced glaucoma. Tools such as HADS, GAD-7, and PSQI should be implemented during routine visits to identify psychiatric comorbidities early and interventions like CBT or support groups may play a key role in alleviating the psychological impact of glaucoma.

This review’s findings highlight the importance of incorporating routine screening for depression, anxiety, and other psychological disorders into standard glaucoma management. Effective collaboration among ophthalmologists, psychiatrists, and primary care providers is essential for designing comprehensive treatment plans that improve both visual and mental health outcomes.

## Figures and Tables

**Figure 1 biomedicines-13-00116-f001:**
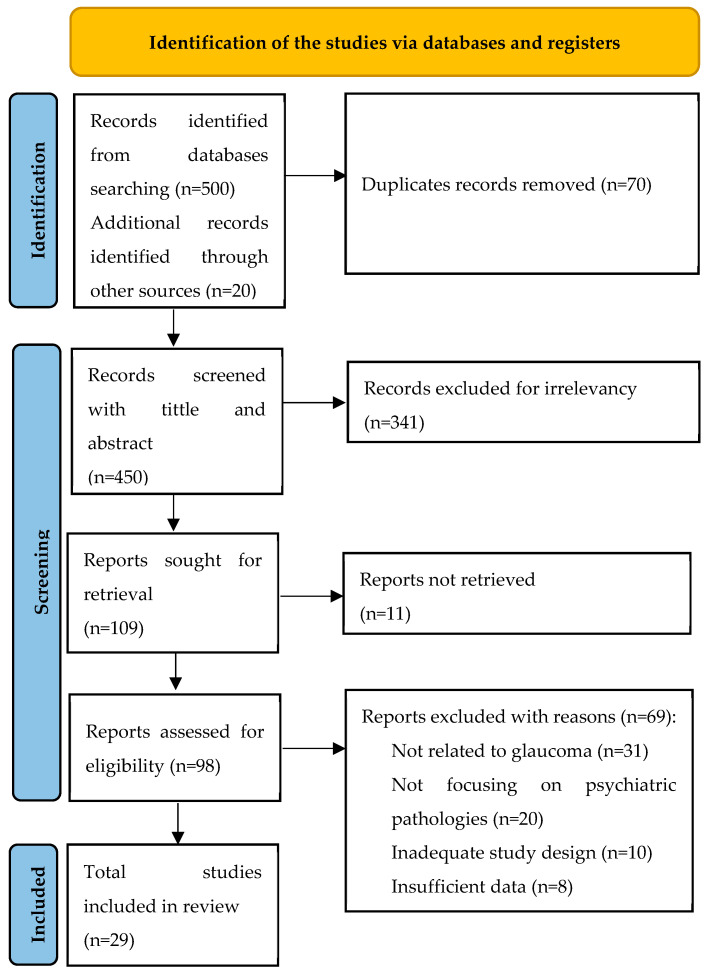
PRISMA model of the literature selection.

**Figure 2 biomedicines-13-00116-f002:**
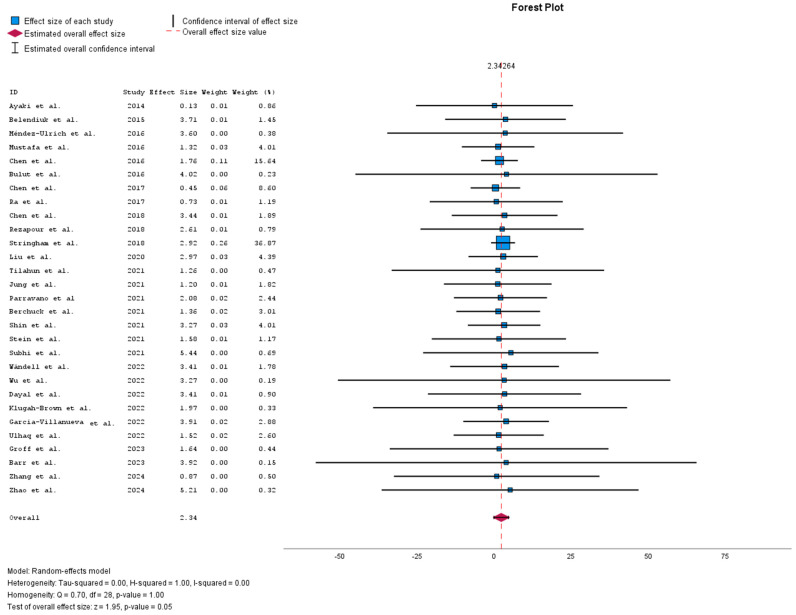
Meta-analysis forest plot [6,9,12,13,14,15,16,17,18,19,20,21,22,23,24,25,26,27,28,29,30,31,32,33,34,35,36,37,38].

**Figure 3 biomedicines-13-00116-f003:**
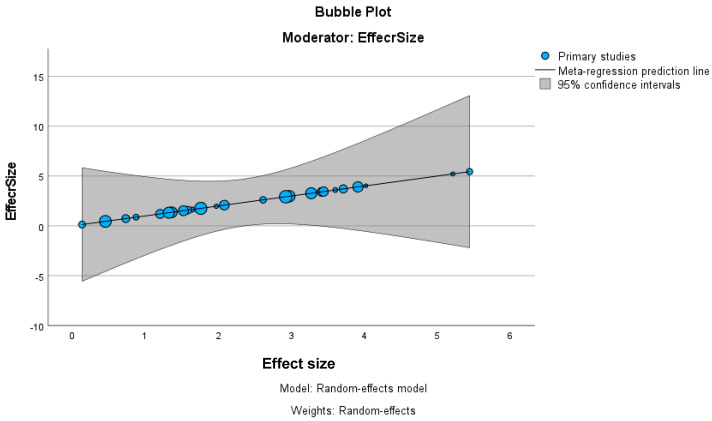
Meta-analysis bubble plot.

**Figure 4 biomedicines-13-00116-f004:**
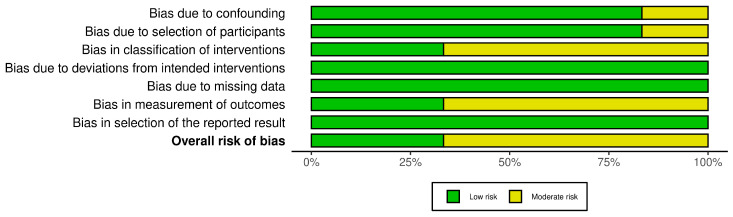
The ROBINS-I tool.

**Figure 5 biomedicines-13-00116-f005:**
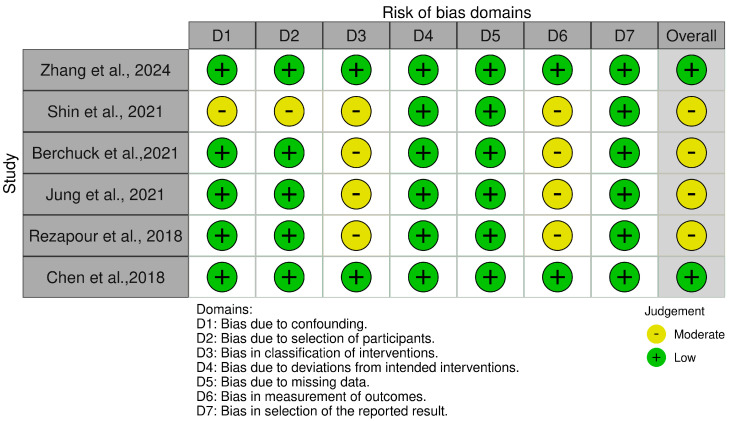
The ROBINS-I traffic light plot [20,21,25,26,27,37].

**Figure 6 biomedicines-13-00116-f006:**
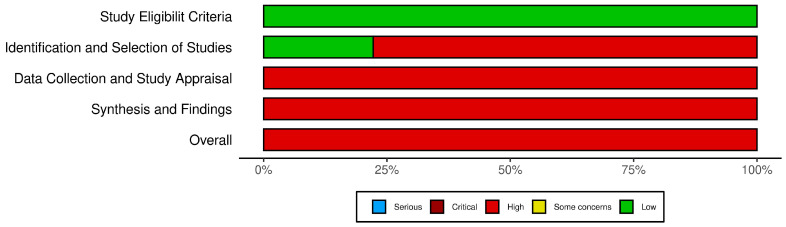
The ROBIS tool.

**Figure 7 biomedicines-13-00116-f007:**
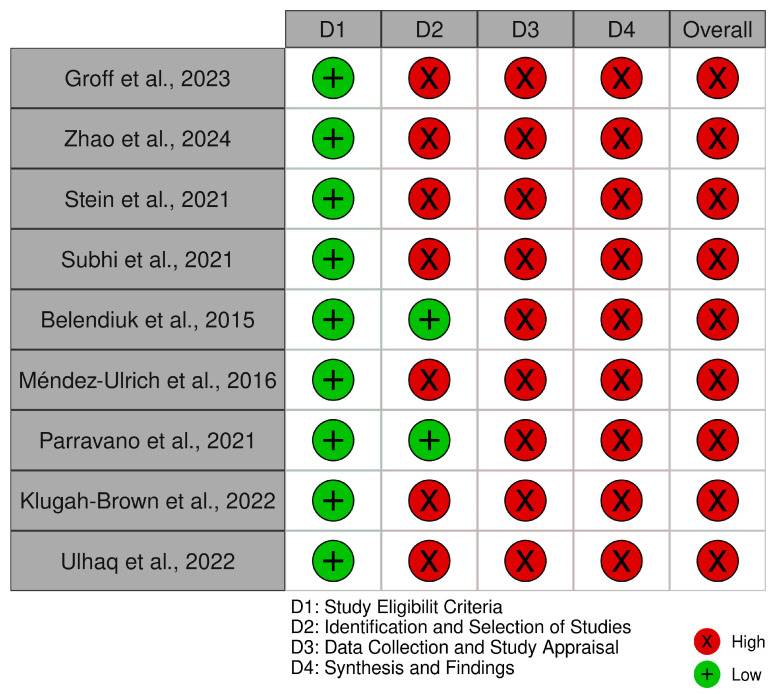
The ROBIS traffic light plot [6,9,13,14,28,29,33,35,38].

**Figure 8 biomedicines-13-00116-f008:**
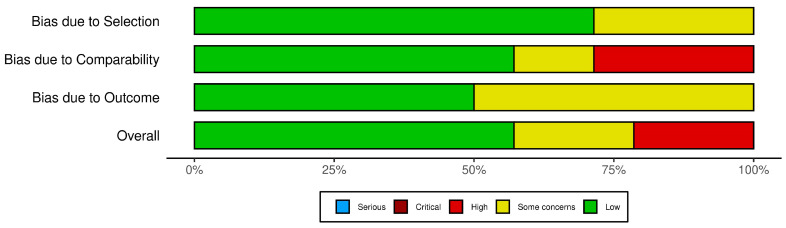
The NOS tool.

**Figure 9 biomedicines-13-00116-f009:**
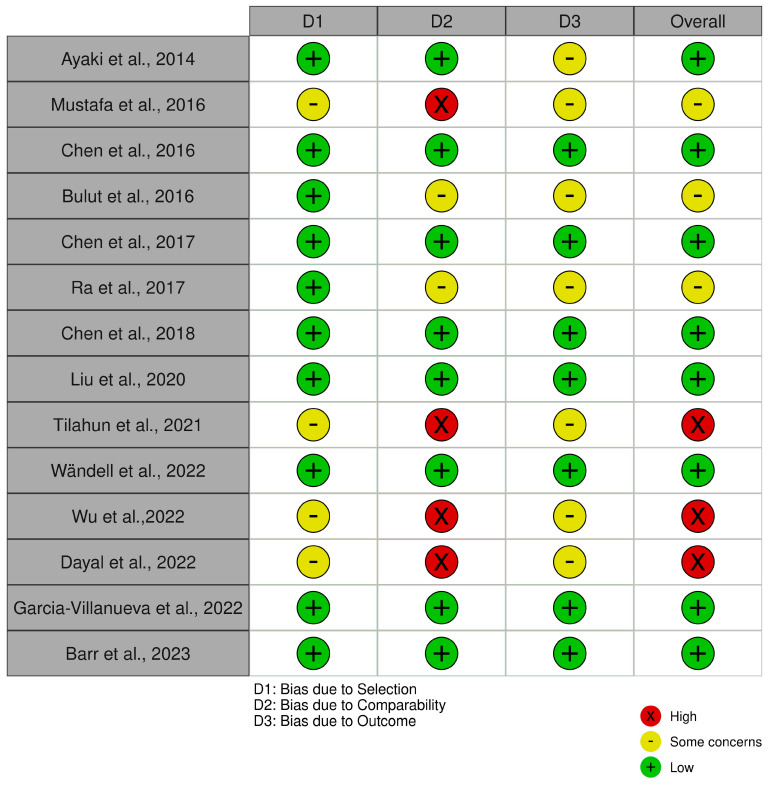
The NOS traffic light plot [12,15,16,17,23,24,30,31,32,34,36].

**Table 1 biomedicines-13-00116-t001:** Data extraction table.

Number	Authors	Year	Country	Study Design	Sample Size	Population Characteristics	Psychiatric Pathologies Examined	Glaucoma Type	Measurement Tools	Results
1	Ayaki et al. [12]	2014	Japan	Cross-sectional case–control study	140 subjects	Glaucoma patients (69) with advanced and moderate glaucoma, normal controls (71)	NR	Advanced and Moderate Glaucoma	Pittsburgh Sleep Quality Index (PSQI), Hospital Anxiety and Depression Scale (HADS), visual field test (Humphrey Visual Field Analyzer), optical coherence tomography (OCT)	Patients with advanced glaucoma had significantly worse PSQI scores than controls (*p* < 0.05)—PSQI correlated with mean deviation in worse eye, number and frequency of medications, and HADS subscores after adjustment for age and sex (*p* < 0.05); no significant correlation between PSQI scores and retinal nerve fiber layer thickness—41% of controls, 58% of advanced, and 49% of moderate glaucoma patients had probable sleep disorders (PSQI > 5)—38% of controls, 57% of advanced, and 42% of moderate glaucoma patients had probable mood disorders (HADS > 9).
2	Belendiuk et al. [13]	2015	USA	Narrative review	Not specified	Patients with various medical and psychological conditions, including glaucoma	post-traumatic stress disorder (PTSD)	NR	Systematic review of literature	Insufficient evidence to support marijuana for PTSD treatment; more rigorous research needed.
3	Méndez-Ulrich et al. [14]	2016	Catalonia, Spain	Literature review	66 studies	Patients with glaucoma	Anxiety, depression	NR	Hospital Anxiety and Depression Scale (HADS), State-Trait Anxiety Inventory (STAI), Beck Depression Inventory (BDI), Minnesota Multiphasic Personality Inventory (MMPI-2)	Glaucoma diagnosis associated with increased anxiety, stress, and depression; prevalence of anxiety and depression higher in glaucoma patients compared to controls; association between personality dimensions and glaucoma.
4	Mustafa et al. [15]	2016	Scotland, UK	Cross-sectional observational study	99 patients	Patients with Normal Tension Glaucoma (NTG)	Cognitive impairment, psychiatric symptoms	NTG	Mini Mental State Examination (MMSE), The auditory verbal learning test (AVLT), General Health Questionnaire (GHQ)	High proportion of patients with impaired MMSE, AVLT, or GHQ scores—28% had significant short-term memory impairment on AVLT—28% scored 5 or more on the GHQ, suggesting psychiatric cases—12% scored 24 or less on the MMSE—30% showed age excessive generalized cerebral atrophy on neuroimaging—20% showed excessive small vessel disease on neuroimaging; common comorbidities included systemic hypertension, ischemic heart disease, diabetes, migraine, transient ischemic attack, and Raynaud’s disease; ophthalmologists should be aware of undiagnosed cognitive impairment and psychiatric morbidity in some patients with NTG.
5	Chen et al. [16]	2016	Taiwan	Case–control study	1465 patients 5712 controls	Ethnic Chinese population in Taiwan with Acute angle-closure glaucoma (AACG)	Selective serotonin reuptake inhibitors (SSRIs)	AACG	Odds ratios (ORs) and 95% confidence intervals (CIs)	Immediate SSRI users had an adjusted OR of 5.80 (95% CI, 1.89–17.9) for AACG compared to nonusers; participants with a mean daily SSRI dose exceeding 20 mg had an adjusted OR of 8.53 (95% CI, 1.65–44.0) for AACG compared to nonusers.
6	Bulut et al. [17]	2016	Brazil	Cross-sectional study	60 participants (20 POAG, 20 NTG, and 20 Control subjects)	Patients with NTG and POAG	Cognitive impairment	POAGNTG	MMSE, spectral domain-optical coherence tomography (SD-OCT	The mean RNFL thicknesses were significantly different among the groups (85.2 ± 14.7, 76.8 ± 10.3, and 91.4 ± 7.7 µm in the POAG, NTG, and C subjects, respectively; *p* < 0.001). The mean GC-IPL thicknesses were 77.5 ± 9.7 µm in the POAG group, 73.4 ± 7.8 µm in the NTG group, and 78.8 ± 3.8 µm in the C group. Differences among the groups were not statistically significant (*p* = 0.085). MMSE scores were 26.1 ± 1.4, 25.7 ± 2.3, and 28.8 ± 0.9 in the POAG, NTG, and C groups, respectively. There were significant differences among the three groups (*p* < 0.001).
7	Chen et al. [18]	2017	Taiwan	Population-based study	15,865 cases 77,014 controls	Nationwide population from National Health Insurance Research Database	SSRIs use associated with glaucoma	NR	National Health Insurance Research Database	SSRI exposure associated with increased glaucoma risk (OR = 1.09). Higher risk with long-term/high-dose SSRI treatment. Effects more pronounced in individuals <65 years, without diabetes, hypertension, or hypercholesterolemia.
8	Ra et al. [19]	2017	Japan	Cross-sectional study	1520 patients (579 males and 941 females)	NR	Anxiety, depression, quality of sleep	NR	Schirmer test, tear break-up time, corneal staining scores, Pittsburgh sleep quality index (PSQI), hospital anxiety and depression score (HADS)	The psychiatric sub-analysis of the control (n = 61, 66.2 years) and glaucoma groups (n = 32, 67.3 years) revealed mean scores of 5.02 ± 3.10 and 5.16 ± 3.46 for PSQI (normal range ≤ 5), 9.47 ± 5.61 and 9.42 ± 7.36 for HADS (normal range ≤ 10), 4.84 ± 3.22 and 4.71 ± 3.45 for anxiety (normal range ≤ 5), and 4.63 ± 3.05 and 4.71 ± 4.40 for depression (normal range ≤ 5), respectively, without statistical significance.
9	Chen et al. [20]	2018	Taiwan	retrospective cohort study	8777 glaucoma patients and 35,108 control subjects	Glaucoma patients from Taiwan	Depression	Open angle glaucoma	Kaplan–Meier curves, Cox regression analysis	Glaucoma patients had a significantly higher cumulative hazard of depression.
10	Rezapour et al. [21]	2018	Europe	Cohort study	14,657 participants	Participants aged 35 to 74 years, 293 with self-reported glaucoma	Depression, Anxiety	Self-reported glaucoma	PHQ-9 for depression, General Anxiety Disorder (GAD)-2 for anxiety	Prevalence of depression: 6.6% (glaucoma) vs. 7.7% (no glaucoma); Prevalence of anxiety: 5.3% (glaucoma) vs. 6.6% (no glaucoma); No significant association between self-reported glaucoma and depression (OR 1.10, *p* = 0.80) or anxiety (OR 1.48, *p* = 0.35).
11	Stringham et al. [22]	2018	USA	Cross-sectional survey study	74 patients	Veterans, patients at Miami VA Eye Clinic	Anxiety disorders, PTSD	NR	63-question survey, DEQ5 score for dry eye symptoms	80% compliance with glaucoma therapy. Dry eye symptoms (DEQ5 score ≥ 6) associated with decreased compliance (63% vs. 89%, *p* = 0.007). Anxiety and PTSD also associated with noncompliance (64% vs. 83%, *p* = 0.05 and 58% vs. 84%, *p* = 0.02). Demographics and depression not significantly associated with compliance.
12	Liu et al. [23]	2020	Taiwan	Retrospective case–control, population-based study	1,000,000 sampled patients	Patients aged ≥20 diagnosed between 1997 and 2013	Schizophrenia, Bipolar Disorder (BD), Major Depressive Disorder (MDD)	Glaucoma suspect Open-angle glaucoma, Closed-angle glaucoma, Age-related macular degeneration, Dry eye syndrome	Retrospective data analysis	In the BD group: glaucoma (OR 1.49), open-angle glaucoma (OR 2.08), closed-angle glaucoma (OR 2.12); in the MDD group: glaucoma (OR 1.24), open-angle glaucoma (OR 1.47), dry eye syndrome (OR 1.22); in the schizophrenia group: glaucoma (OR 1.47), glaucoma suspect (OR 1.88), open-angle glaucoma (OR 2.19).
13	Tilahun et al. [24]	2021	Ethiopia	Cross-sectional study	495 glaucoma patients	Adults with glaucoma	Common mental disorders (CMD)	Self-reported glaucoma	Self-Reporting Questionnaire (SRQ-20)	Prevalence of CMD: 29.5% (95% CI 25.4–33.3).
14	Jung et al. [25]	2021	South Korea	Cohort study	922,769 subjects	Subjects aged 66 receiving National Screening Program during 2009–2014. Includes both genders	Depression	NR	National Health Insurance Database (KNHID), National Screening Program for Transitional Ages (NSPTA)	Subjects with depression showed increased risk of developing glaucoma.
15	Parravano et al. [6]	2021	Europe, USA, Asia	Meta-analysis	6992 subjects	Patients with visual impairment, mean age 76 (SD 13.9), 60% women	Depression	NR	Random-intercept logistic regression models, PRISMA guidelines	Among 6992 patients with visual impairment, the median proportion of depression was 0.30 (range, 0.03–0.54). Random-effects pooled estimate was 0.25 (95% CI, 0.19–0.33) with high heterogeneity.
16	Berchuck et al. [26]	2021	USA	Retrospective cohort study	3259 patients	Adults diagnosed as glaucoma suspects at baseline from the Duke Glaucoma Registry. Mean age: 60.0 years, 58% female, 57% Caucasian/White, 34% African American/Black	Anxiety, depression	POAG	Electronic health records (EHR) billing codes, medical history, problem list, Goldmann applanation tonometry, Tono-Pen, Humphrey Field Analyzer (HFA)	911 (28%) developed glaucoma during follow-up. Anxiety and both anxiety and depression were associated with higher risk of progressing to glaucoma (adjusted HRs of 1.16 and 1.27, respectively). Female and Caucasian/White patients had an increased risk of psychiatric diagnosis at baseline. Worse baselines mean deviation (MD) levels were associated with psychiatric diagnosis.
17	Shin et al. [27]	2021	South Korea	Retrospective case–control study	251 eyes with open-angle glaucoma	Glaucoma patients followed for at least 2 years	Anxiety, depression	Open-angle glaucoma	Beck Anxiety Inventory (BAI), Beck Depressive Inventory-II (BDI-II), intraocular pressure (IOP) measurement (Goldmann applanation tonometry), RNFL thickness measurement (Cirrus OCT), Humphrey visual fields (VFs)	Disk hemorrhage, peak IOP, and retinal nerve fiber layer (RNFL) thickness loss rate were significantly associated with high anxiety. Multivariate analysis showed RNFL thinning rate and disk hemorrhage to be significant factors associated with anxiety. Rate of RNFL thickness loss positively correlated with BAI score and IOP fluctuation. Visual field mean deviation and heart rate variability were significantly associated with high depression. Multivariate analysis showed the MD of visual field and heart rate variability to be significant factors associated with depression.
18	Stein et al. [28]	2021	USA	Review	NR	CBS adults with glaucoma, risk factors include older age, non-White race, family history, and certain medications and systemic conditions	Headache	Open-angle glaucoma, angle-closure glaucoma	Intraocular pressure measurement, perimetry, optical coherence tomography	Individuals with glaucoma can experience higher rates of headache than controls
19	Subhi et al. [29]	2021	Sweden	Systematic review and meta-analysis	827 patients	Patients with glaucoma, predominantly elderly, with varying stages and ocular comorbidities	Corticobasal syndrome	NR	Literature search across multiple databases, qualitative and quantitative analysis	Prevalence of Corticobasal syndrome in glaucoma patients with various stages and ocular comorbidities: 2.8% (CI95%: 0.7–6.1%). Prevalence in bilateral low visual acuity patients: 13.5% (CI95%).
20	Wändell et al. [30]	2022	Sweden	Cross-sectional study	1,703,675 individuals	All individuals over 18 years old residing in Stockholm County as of 1 January 2017	Dementia, psychosis, depression	POAG	Regional healthcare database (VAL) data	Lower risk of POAG in men with dementia (OR: 0.714) and women with dementia (OR: 0.653); lower risk of POAG in women with psychosis (OR: 0.478); increased risk of POAG in women with depression (OR: 1.164); higher risk of POAG associated with higher neighborhood socio-economic status
21	Wu et al. [31]	2022	China	Cross-sectional study	446 patients	Chinese patients with glaucoma	Anxiety, depression	POAG, PACG	HADS, NEI VFQ-25, ophthalmological examinations	Prevalence of anxiety (12.11%) and depression (25.78%) in Chinese patients with glaucoma. VR-QoL negatively associated with anxiety and depression; objective visual function indices less indicative of psychological burden.
22	Dayal et al. [32]	2022	India	Cross-sectional prevalence study	200 patients	Glaucoma patients attending a tertiary care eye hospital in Pune, India	Anxiety, depression	Primary glaucoma	HADS questionnaire	Mean HADS-Anxiety score: 4.5 (SD = 3.4). Mean HADS-Depression score: 4.1 (SD = 3.8). Severity of glaucoma associated with significantly higher HADS scores. A third of patients (34.5%) were symptomatic for anxiety or depression, with 12.5% classified as definite cases. The duration of treatment had no association with HADS scores.
23	Klugah-Brown et al. [33]	2022	Various (study used meta-analysis)	Meta-analysis of case–control studies	Glaucoma (207 patients, 174 controls), GAD (226 patients, 226 controls), MDD (2575 patients, 2866 controls)	Glaucoma patients (POAG, NTG, PACG), GAD, MDD	NR	Various types (POAG, NTG, PACG)	MRI, Voxel-based morphometry (VBM)	Glaucoma patients showed gray matter volume reductions in the lingual gyrus, thalamus, putamen, and insula. Reductions in the insular region overlapped with changes observed in GAD. Tripartite brain model of glaucoma proposed, with changes in visual processing regions and additional alterations in putamen and insula associated with motivational and emotional functions. Findings suggest broad neuroanatomical alterations in glaucoma extending beyond the visual system.
24	Garcia-Villanueva et al. [34]	2022	Spain and Portugal	Multicenter, analytical, observational, case–control study	412 participants	Participants aged 40–80 years with Ocular Hypertension (OHT) or OAG; Demographic, epidemiological, ocular/systemic clinical data recorded	Depression, anxiety, sleep disorders	OHT and Open-Angle Glaucoma (OAG)	Systematized ocular examination, IOP measurements, CCT, OCT, VF evaluation, and clinical history	Mean IOP for OHT: 20.46 ± 2.35 mmHg (RE), 20.1 ± 2.73 mmHg (LE); for OAG: 15.8 ± 3.83 mmHg (RE), 16.94 ± 3.86 mmHg (LE). Significant differences in IOP, OCT parameters, and VF mean deviation between OHT and OAG groups. Highest prevalence of overweight/obesity and daily coffee consumption in both groups. Major risk factors for conversion from OHT to OAG: overweight/obesity, migraine, asthma, smoking. Need for increased understanding of non-ocular factors influencing O
25	Ulhaq et al. [35]	2022	Indonesia	Systematic review and meta-analysis	95 studies	Ophthalmic disease patients, including pediatric and adult patients	Anxiety symptoms and disorders	Glaucoma (general)	Random-effects model, data from PubMed, Scopus, Web of Science	Pooled prevalence: 31.2% with anxiety symptoms, 19.0% with anxiety disorders. Pediatric patients had higher anxiety (58.6%) than adults (29%). Anxiety symptoms most prevalent in uveitis (53.5%); 30.7% of glaucoma patients had anxiety symptoms, 22.2% had anxiety disorders. Significant increase in anxiety compared to healthy controls.
26	Groff et al. [9]	2023	Canada	Systematic review and meta-analysis	4,995,538 subjects from 45 studies	Patients with glaucoma	Depression, anxiety, sleep disorders	POAG	Modified Downs and Black checklist	Higher prevalence and severity of depression, anxiety, and sleep disorders.
27	Barr et al. [36]	2023	USA	Genetic association study	230 patients	Average age 78, 38% male, White Non-Hispanic	NR	General glaucoma	Medication Possession Ratio (MPR), Proportion of Days Covered (PDC), Illumina HumanCoreExome BeadChip, Exome Sequencing	59% non-adherence by MPR80, 67% by PDC80.—57% (MPR80) and 48% (PDC80) non-adherence attributed to genetic components. Significant missense mutations found in *TTC28, KIAA1731, ADAMTS5, OR2W3, OR10A6, SAXO2, KCTD18, CHCHD6, UPK1A*, *TINAG, GSTZ1*, and *SEMA4G*. SNP rs6474264 within ZMAT4 nominally significant (*p* = 5.54 × 10^−6^) with decreased non-adherence risk (OR, 0.22). *CHCHD6* linked to Alzheimer’s disease increased non-adherence risk by three-fold (95% CI, 1.62–5.8). Non-adherence higher in men (72% PDC80, 64% MPR80) compared to women (62% PDC80, 56% MPR80). The age group of 40–64 showed higher non-adherence (79% PDC80, 68% MPR80) compared to 65 and over (65% PDC80, 58% MPR80).
28	Zhang et al. [37]	2024	China	Mendelian randomization study	N/A	European and East Asian populations	Depression, insomnia, schizophrenia	POAG PACG	Two-sample MR analysis, GWAS data, IVW method, heterogeneity, pleiotropy, Steiger directionality test	No causal relationship between psychiatric disorders (depression, insomnia, schizophrenia) and glaucoma in both European and East Asian populations. OR and CI values indicated no significant association. Results were consistent and robust across different populations and methods.
29	Zhao et al. [38]	2024	China	Meta-analysis of cohort studies	11 studies, 4,645,925 participants	Adult population	Alzheimer’s disease (AD), dementia, non-AD dementia	POAG PACG	Random-effects model, systematic search in PubMed, Embase, Cochrane’s Library	No independent association between glaucoma and increased incidence of AD (adjusted RR: 1.03, 95% CI: 0.93–1.05), all-cause dementia (adjusted RR: 1.08, 95% CI: 0.97–1.19), or non-AD dementia (adjusted RR: 1.05, 95% CI: 0.91–1.21). Consistent results across subgroups.

NR—Not reported.

**Table 2 biomedicines-13-00116-t002:** Trends in psychiatric comorbidities among glaucoma patients.

Category	Key Findings	Discrepancies/Insights
**Geographic Trends**	Asia: Studies from Japan (Ayaki et al., 2014) [12] and Taiwan (Chen et al., 2017) [18] report significant anxiety and depression prevalence. Taiwan: higher depression prevalence among glaucoma patients on SSRIs (adjusted OR 5.80 for angle-closure glaucoma). Japan: advanced glaucoma linked to worse sleep quality (PSQI) and higher mood disorder rates.	Asian populations report lower depression prevalence compared to Western countries (e.g., 12.11% in China vs. up to 57% in Western populations), possibly due to cultural stigma or differences in self-reporting norms (Wu et al., 2022) [31].
	Western Countries (Europe and North America): Higher rates of depression and anxiety. USA (Groff et al., 2023) [9]: Pooled anxiety prevalence 31.2%, depression 19.0%. Europe (Rezapour et al., 2018) [21]: Similar trends, with variations influenced by healthcare access and diagnostic tools. Psychiatric medication use (e.g., SSRIs) often linked to glaucoma progression (Chen et al., 2017) [18].	Western countries emphasize pharmacological factors and have better access to mental health care, possibly inflating diagnosis rates compared to regions with limited resources.
	Africa: Ethiopia (Tilahun et al., 2021) [24]: CMD prevalence of 29.5%, higher than many Western studies, due to limited access to glaucoma management and mental health services.	African studies often focus on systemic barriers to care (poverty, limited healthcare) rather than neurobiological factors, contrasting with high-income countries where pharmacological and diagnostic factors are emphasized.
**Cultural Trends**	Stigma and Mental Health Awareness: Asian and African populations underreport psychiatric symptoms due to cultural stigma. Depression prevalence: 25.78% in Chinese glaucoma patients (Wu et al., 2022) [31] vs. up to 57% in Western studies.	Cultural differences influence reporting rates: Asian and African cultures focus on somatic complaints, while Western cultures emphasize psychological symptoms.
	Perception of Glaucoma: Western countries treat glaucoma as a chronic disease, while in lower-income regions, it is seen as life-altering, leading to feelings of helplessness.	Cultural differences in disease perception influence mental health outcomes. In lower-income regions, psychological distress may be heightened due to limited resources and poor disease education.
	Social Support Systems: Strong family structures in Asian and African cultures mitigate glaucoma’s psychological impact. The Western emphasis on individualism may increase feelings of isolation.	Social support can buffer mental health impacts in collectivist cultures but may be weaker in individualistic societies, worsening isolation and psychological distress.
**Clinical Trends**	Advanced vs. Early-Stage Glaucoma: Advanced glaucoma universally associated with worse psychiatric outcomes. Japan (Ayaki et al., 2014) [12]: poor sleep quality correlates with advanced disease. USA (Berchuck et al., 2021) [26]: disease progression increases anxiety and depression risks.	Younger patients with early-stage glaucoma report higher anxiety levels, likely due to concerns about managing lifelong disease (Shin et al., 2021) [27].
	Special Populations (Pediatric Glaucoma): Pediatric patients have significantly higher anxiety levels (58.6%) compared to adults (29%), reflecting unique psychological challenges (Ulhaq et al., 2024) [35].	Children may require tailored mental health interventions, as their psychological burden differs from adults and may stem from parental concerns and long-term prognosis fears.
	Sleep Disorders: Sleep disturbances linked to disease severity (Groff et al., 2023; Ayaki et al., 2014) [9,12]. Stress and discomfort in advanced glaucoma exacerbate psychiatric symptoms.	Sleep disturbances occur across regions but are more frequently reported in severe cases due to stress and optic nerve damage.
**Socioeconomic and Healthcare Access Trends**	Higher Socioeconomic Status (SES): Linked to increased depression and anxiety in some studies (e.g., Wändell et al., 2022) [30], potentially due to better access to diagnostic services and heightened health awareness.	Paradoxical association of higher SES with psychiatric distress may reflect overdiagnosis or greater health-related anxiety.
	Resource-Limited Settings: Untreated glaucoma in low-income regions leads to higher psychological distress due to limited access to care and social stigma (Tilahun et al., 2021) [24].	Resource constraints exacerbate the mental health burden in LMICs, underscoring the need for accessible healthcare systems and public awareness campaigns.

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
