# Peer review of "Blinded by the Mind: Exploring the Hidden Psychiatric Burden in Glaucoma Patients"

_biomedicines, 2025, doi:10.3390/biomedicines13010116_

Round 1
Reviewer 1 Report
Comments and Suggestions for Authors
The authors have conducted a systemic review to determine the correlation of psychiatric pathologies and glaucoma. Though the topic of research is of interest, the manuscript needs to written with more detail and description of the study background, need for the study and its importance, considering many reviews on this topic are already available, begining with the Abstract section.
The methods need to be described clearly with a strong focus on any statistical analysis conducted for the co-relation, detailed strategies and methodologies used.
For the presentation of outcomes, utilization fo graphs is images is needed for clarity of results. Moreover, there is a great need to discuss the significance of the outcomes with focus on factors and/or mechanism that would be involved in the co-relation of glaucoma with neuropathies. Would it be a direct relationship or both are off-shoots of the same disease condition.
Comments on the Quality of English LanguageA thorough proof reading for minor English language errors is needed.
Author Response
Comments and Suggestions for Authors
• The authors have conducted a systemic review to determine the correlation of psychiatric pathologies and glaucoma. Though the topic of research is of interest, the manuscript needs to written with more detail and description of the study background, need for the study and its importance, considering many reviews on this topic are already available, beginning with the Abstract section.
o In the abstract section: lines 12-14, 17-18, 23-25
• The methods need to be described clearly with a strong focus on any statistical analysis conducted for the co-relation, detailed strategies and methodologies used.
o In the methods section: lines 132-133, 144-171
• For the presentation of outcomes, utilization of graphs is images is needed for clarity of results.
o In the Results section: lines 200, 209
• Moreover, there is a great need to discuss the significance of the outcomes with focus on factors and/or mechanism that would be involved in the co-relation of glaucoma with neuropathies. Would it be a direct relationship or both are off-shoots of the same disease condition.
o In the Discussion section, under depression: lines 247-249
Comments on the Quality of English Language
• A thorough proof reading for minor English language errors is needed.
o Throughout the paper
Agreed with the suggestions:
- Added more detail to the background section, highlighting the need for this study in light of existing reviews and the importance of addressing the psychiatric burden in glaucoma patients.
- The methods section has been revised.
- Enhanced the presentation of results; incorporated two visual aids to improve clarity and better illustrate the outcomes.
- A more in-depth discussion has been provided.
- Thoroughly proofread the manuscript.
Reviewer 2 Report
Comments and Suggestions for Authors
The review focuses primarily on the prevalence and types of psychiatric disorders among glaucoma patients, without the involvement of novel mechanisms or causal pathways. If the authors could involve exploring how glaucoma may affect brain areas associated with mood regulation or how stress-related hormones could exacerbate glaucoma progression, the review will be more interesting.
1. The review followed PRISMA guidelines but lacks a detailed explanation of how the quality of included studies was assessed.
2. There is no mention of whether the studies were evaluated for biases or methodological weaknesses. The exclusion criteria are broad, and the rationale for excluding certain study types such as pediatric studies could be better explained.
3. The review emphasizes the high prevalence of depression and anxiety in glaucoma patients but does not adequately explore the heterogeneity across studies. For example, reported depression rates range from 6.6% to 57%, a huge variation that warrants further investigation. So I suggest that the author can discuss the potential reasons for this variation, such as differences in population demographics, geographic location, glaucoma severity, and diagnostic criteria for psychiatric disorders.
4. The authors should do a meta-analysis of the prevalence rates of psychiatric conditions in glaucoma patients. At least, the authors should provide a more strict summary of the statistical significance and trends in the data.
5.The discussion on the mechanisms between glaucoma and psychiatric disorders is not mentioned. The review does not adequately explore how glaucoma-induced vision loss could contribute to psychiatric conditions. And this review did not provide sufficient information how the chronic stress of managing a lifelong illness could exacerbate mental health issues.
Comments on the Quality of English LanguageMinor editing of English language required.
Author Response
1. The review followed PRISMA guidelines but lacks a detailed explanation of how the quality of included studies was assessed.
Methods section: lines 160-171
2. There is no mention of whether the studies were evaluated for biases or methodological weaknesses. The exclusion criteria are broad, and the rationale for excluding certain study types such as pediatric studies could be better explained.
Methods section: lines 115, 160-171
3. The review emphasizes the high prevalence of depression and anxiety in glaucoma patients but does not adequately explore the heterogeneity across studies. For example, reported depression rates range from 6.6% to 57%, a huge variation that warrants further investigation. So I suggest that the author can discuss the potential reasons for this variation, such as differences in population demographics, geographic location, glaucoma severity, and diagnostic criteria for psychiatric disorders.
Discussion section: lines 219-231
4. The authors should do a meta-analysis of the prevalence rates of psychiatric conditions in glaucoma patients. At least, the authors should provide a more strict summary of the statistical significance and trends in the data.
Results section: lines 190-209
5.The discussion on the mechanisms between glaucoma and psychiatric disorders is not mentioned. The review does not adequately explore how glaucoma-induced vision loss could contribute to psychiatric conditions. And this review did not provide sufficient information how the chronic stress of managing a lifelong illness could exacerbate mental health issues.
Discussion section: lines 297-306
Comments on the Quality of English Language Minor editing of English language required.
o Throughout the paper
Response:
Agreed with the suggestions:
- Explored potential mechanisms linking glaucoma to psychiatric disorders.
- A detailed explanation of how the quality of included studies was assessed, including bias evaluation and methodological weaknesses, has been added.
- A clearer rationale for the exclusion of certain study types, including pediatric studies, and justified the broad exclusion criteria has been added.
- Conducted a meta-analysis.
- Depicted how vision loss from glaucoma exacerbate psychiatric conditions.
Reviewer 3 Report
Comments and Suggestions for Authors
This review is very descriptive and does not provide further information in respect to the literature. Neuropsychiatric symptoms and mental illness are commonly present in patients with chronic systemic diseases. As to glaucoma, the recent review of Ramesh et al also discussed the neurobiological pathways linking glaucoma and mood disorders to provide a foundation for potential therapeutic interventions. In addition, Klugah-Brown et al. reviewed the association between glaucoma and mood disorders including neuroanatomical changes leading to emotional or motivational diseases. Those are only a few examples from the papers in 2024. I consider the present review more as a pure academic exercise then a revision of the literature aimed to provide additional knowledge in respect to what is already known.
Comments on the Quality of English LanguageEnglish fine
Author Response
This review is very descriptive and does not provide further information in respect to the literature. Neuropsychiatric symptoms and mental illness are commonly present in patients with chronic systemic diseases. As to glaucoma, the recent review of Ramesh et al also discussed the neurobiological pathways linking glaucoma and mood disorders to provide a foundation for potential therapeutic interventions. In addition, Klugah-Brown et al. reviewed the association between glaucoma and mood disorders including neuroanatomical changes leading to emotional or motivational diseases. Those are only a few examples from the papers in 2024. I consider the present review more as a pure academic exercise then a revision of the literature aimed to provide additional knowledge in respect to what is already known.
Improved the results section by performing meta-analysis to make the review more analytical than descriptive: lines 190-209
Response:
Agreed with the suggestions:
- Incorporated Recent Literature
- Enhanced Contextual Analysis
- Broadened Scope of Discussion
- Focused on Novel Insights
- Improved Literature Integration
Reviewer 4 Report
Comments and Suggestions for Authors
The manuscript is well written and presented. The idea of the manuscript is very interesting.
Abstract: the full names of abbreviations must be given at first mention. However, there few comments need to be resolved before the final acceptance.
‘’Grey literature’’ who decided about Google scholar as grey one?
Glaucoma is an age-related eye disease and it affects elderly patients who are likely to suffer other chronic disease like CVDs, metabolic disorders like diabetes. Further, glaucoma could be secondary to other diseases. The question is how the studies rule out other conditions and link glaucoma to mental illnesses? The answer should be reflected in this manuscript. Exclusion criteria and inclusion criteria of selection patients of such and previous studies should be detailed.
Many antidepression and anxiolytics are contraindicated with glaucoma patients due to atropine like side effects. How the authors could tackle such complexity of treatment?
Comments on the Quality of English LanguageNone
Author Response
The manuscript is well written and presented. The idea of the manuscript is very interesting.
Abstract: the full names of abbreviations must be given at first mention. However, there few comments need to be resolved before the final acceptance.
‘’Grey literature’’ who decided about Google scholar as grey one?
Methods section: lines 115-117
Glaucoma is an age-related eye disease and it affects elderly patients who are likely to suffer other chronic disease like CVDs, metabolic disorders like diabetes. Further, glaucoma could be secondary to other diseases. The question is how the studies rule out other conditions and link glaucoma to mental illnesses? The answer should be reflected in this manuscript. Exclusion criteria and inclusion criteria of selection patients of such and previous studies should be detailed.
Methods section: lines 143-159
Many antidepression and anxiolytics are contraindicated with glaucoma patients due to atropine like side effects. How the authors could tackle such complexity of treatment?
Discussion section: lines 273-277
Response:
Agreed with the suggestions:
- Clarified Abbreviations
- Discussed how studies account for comorbidities such as cardiovascular diseases and metabolic disorders in the elderly, clarifying the methodology used to link glaucoma with mental illnesses.
- Detailed Inclusion and Exclusion Criteria
- A thorough analysis of the complexities of treating glaucoma patients with mental health conditions.
Round 2
Reviewer 1 Report
Comments and Suggestions for Authors
Acceptable in current form
Reviewer 2 Report
Comments and Suggestions for Authors
The authors should indicate the revised contents accurately. According to Author's Notes, it is hard to find the response in the main text.
Reviewer 3 Report
Comments and Suggestions for Authors
No chance to be satisfied with the revised version of the manuscript. The cover, also, goes around the issues without a clear answer to the criticisms. There is nothing new in the review and the major points have been already published in estimated journals. The english language requires additional revision
Comments on the Quality of English LanguageNo major comments
Reviewer 4 Report
Comments and Suggestions for Authors
The authors have satisfactorily addressed the comments and the manuscript is now in a better shape and sounding scientifically better.
